

# Bioinformatic prediction of the molecular links between Alzheimer's disease and diabetes mellitus

Ricardo Castillo-Velázquez[1,2], Flavio Martínez-Morales[3], Julio E. Castañeda-Delgado[1,4], Mariana H. García-Hernández[1], Verónica Herrera-Mayorga[5], Francisco A. Paredes-Sánchez[5], Gildardo Rivera[6], Bruno Rivas-Santiago[1] and Edgar E. Lara-Ramírez[1,6]

[1] Unidad de Investigación Biomédica de Zacatecas, Instituto Mexicano del Seguro Social, Zacatecas, Zacatecas, México
[2] Centro de Investigación en Ciencias de la Salud y Biomedicina, Universidad Autónoma de San Luis, San Luis Potosí, San Luis Potosí, México
[3] Departamento de Farmacología, Facultad de Medicina, Universidad Autónoma de San Luis, San Luis Potosí, San Luis Potosí, México
[4] Investigadores por México, CONACYT, Consejo Nacional de Ciencia y Tecnología, Zacatecas, Zacatecas, México
[5] Unidad Académica Multidisciplinaria Mante, Universidad Autónoma de Tamaulipas, Mante, Tamaulipas, México
[6] Laboratorio de Biotecnología Farmacéutica, Centro de Biotecnología Genómica, Instituto Politécnico Nacional, Reynosa, Tamaulipas, México

Corresponding author
Edgar E. Lara-Ramírez,
doc_lara_ram@hotmail.com

## ABSTRACT

**Background.** Alzheimer's disease (AD) and type 2 diabetes mellitus (DM2) are chronic degenerative diseases with complex molecular processes that are potentially interconnected. The aim of this work was to predict the potential molecular links between AD and DM2 from different sources of biological information.

**Materials and Methods.** In this work, data mining of nine databases (DisGeNET, Ensembl, OMIM, Protein Data Bank, The Human Protein Atlas, UniProt, Gene Expression Omnibus, Human Cell Atlas, and PubMed) was performed to identify gene and protein information that was shared in AD and DM2. Next, the information was mapped to human protein-protein interaction (PPI) networks based on experimental data using the STRING web platform. Then, gene ontology biological process (GOBP) and pathway analyses with EnrichR showed its specific and shared biological process and pathway deregulations. Finally, potential biomarkers and drug targets were predicted with the Metascape platform.

**Results.** A total of 1,551 genes shared in AD and DM2 were identified. The highest average degree of nodes within the PPI was for DM2 (average = 2.97), followed by AD (average degree = 2.35). GOBP for AD was related to specific transcriptional and translation genetic terms occurring in neurons cells. The GOBP and pathway information for the association AD-DM2 were linked mainly to bioenergetics and cytokine signaling. Within the AD-DM2 association, 10 hub proteins were identified, seven of which were predicted to be present in plasma and exhibit pharmacological interaction with monoclonal antibodies in use, anticancer drugs, and flavonoid derivatives.
**Conclusion**. Our data mining and analysis strategy showed that there are a plenty of biological information based on experiments that links AD and DM2, which could provide a rational guide to design further diagnosis and treatment for AD and DM2.

## INTRODUCTION

Alzheimer's disease (AD) and type 2 diabetes mellitus (DM2) are chronic degenerative human diseases with complex molecular processes (*Chornenkyy et al., 2019*). Worldwide, AD affects around 3% of the population with an age range of 65 to 74 years, while DM2 is the fastest growing metabolic disease in the world in adults older than 50 years (*Chen, Magliano & Zimmet, 2012*; *Mayeux & Stern, 2012*). Patients with DM2 have been shown to be around 50% more likely to have a decrease in cognitive ability, leading to dementia where AD is the most common cause (*Ryan, Fine & Rosano, 2014*; *Moheet, Mangia & Seaquist, 2015*).

The relationship of DM2 with AD has attracted the attention of the scientific community due to its possible link derived from epidemiological research (*Curb et al., 1999*). Other studies showed that the main relationship between these two diseases is a process of insulin resistance in the brain (*Nicolls, 2004*; *Arnold et al., 2018*), which has led to the proposed association known as "type 3 diabetes mellitus", (DM3) (*Kandimalla, Thirumala & Reddy, 2017*). In fact, evidence showed that patients with DM2 may have up to 3 times the risk of suffering from AD compared to people without DM2 (*Li, Song & Leng, 2015*).

To date, several research groups are working to understand how these two diseases are connected. Experimental research has produced valuable information on how AD and DM2 could be linked. *Liu et al. (2011)* found that the levels and activity of various components of Insulin—phosphoinositide 3 kinase (PI3K)—AKT serine/threonine kinase pathway decreased in cases of AD and DM2, and the decrease in this pathway is more serious in cases of AD-DM2 association than in DM2 or AD alone. The use of systems biology tools such as protein-protein interactions (PPI) predictions could provide valuable information on biological processes shared between different diseases (*Sharan & Ideker, 2006*). PPI data can now be extracted from repositories or databases and analyzed to obtain new information on the functions or relationships of proteins using bioinformatic tools, which can thus be used to make new predictions of signaling networks on a large scale (*Stelzl et al., 2005*). This approach has been used to study the AD and DM2 association (*Mittal, Mani & Katare, 2016*). In this study were identified shared cellular and molecular mechanisms alterations such as beta ($\beta$) cell development, negative regulation of PI3K/AKT signaling pathway, β-amyloid and insulin degradation. Other studies have used these tools to identify new genes and possible pharmacological targets in AD (*Rahman et al., 2019a*). Similarly, the application of these tools has been carried out in other related diseases such as type 1 diabetes mellitus (DM1). The results have led to the identification of genes involved

in important biological processes in this disease DM1 (*Chen et al., 2021*), or even in other related diseases such as heart disease and related complications in diabetic patients (*Kumar et al., 2020*).

In the present study, several databases were mined to identify molecular information shared by AD and DM2. The obtained information was analyzed with the help of gene ontology biological process (GOBP), signaling pathways, and hub proteins that potentially connect both diseases. The molecular interactions described here could contribute to the elucidation of the pathophysiological processes underlying the AD—DM2 association. Furthermore, potential biomarkers and drug targets for its diagnosis and treatment were identified.

## MATERIALS & METHODS

### Gene and protein data mining

The workflow for gene and protein data mining is shown in Fig. 1. Genes and proteins related to AD and DM2 were obtained from nine databases: DisGeNET (https://www.disgenet.org/), Ensembl (https://www.ensembl.org/index.html), OMIM (https://omim.org/), Protein Data Bank (https://www.rcsb.org/), The Human Protein Atlas (http://www.proteinatlas.org/), UniProt (https://www.uniprot.org/), Human Cell Atlas (https://www.humancellatlas.org/), PubMed (https://pubmed.ncbi.nlm.nih.gov/), and Gene Expression Omnibus (GEO, https://www.ncbi.nlm.nih.gov/geo/). The key words "AD", "DM2", "Alzheimer", "type 2 Diabetes Mellitus" were used to search on each database. Further data processing was performed for the following databases: The PubMed gene information (https://pubmed.ncbi.nlm.nih.gov/) was retrieved with the PubTator application (https://www.ncbi.nlm.nih.gov/research/pubtator/) (*Wei et al., 2019*) and the related PMID was recorded in our local database. In the Human Cell Atlas, the genes determined in single-cell sequencing on AD peripheral blood (*Xu & Jia, 2021*), with a log2 Fold Change (FC) $\geq 1$, and adjusted $p$-values $<0.05$ applying the Benjamini–Hochberg correction (BHC) were considered. In GEO, the data sets GSE5281, GSE122063 were chosen for AD, and the GSE7014, GSE29221 for DM2. Those files comprise 297 patients for AD and 60 for DM2; the differentially expressed genes (DEGs) were determined with the GEO2R tool (https://www.ncbi.nlm.nih.gov/geo/geo2r/) (*Barrett et al., 2013*). DEG selection was also based on log2 FC $\geq 1$ and adjusted $p$-values $<0.05$ using the BHC method. The information obtained was organized into the gene and protein lists according to the disease information. These lists were curated by eliminating duplicates and synonymous names with the help of the HUGO Gene Nomenclature Committee (http://www.genenames.org, accessed on January 2022) (*Bruford et al., 2021*). The final lists were compared with the help of Venn diagrams (http://bioinformatics.psb.ugent.be/beg/) to identify the genes shared between AD and DM2. The intersected information was considered as the AD-DM2 association.

### Generation of protein-protein interactions (PPI) networks

PPI manages important biological processes (*Rao et al., 2014*). To produce PPI for each study group, the gene lists were submitted to the STRING platform (STRING; v11.0; http://string-db.org) to determine PPI in humans. The parameters to predict PPI were

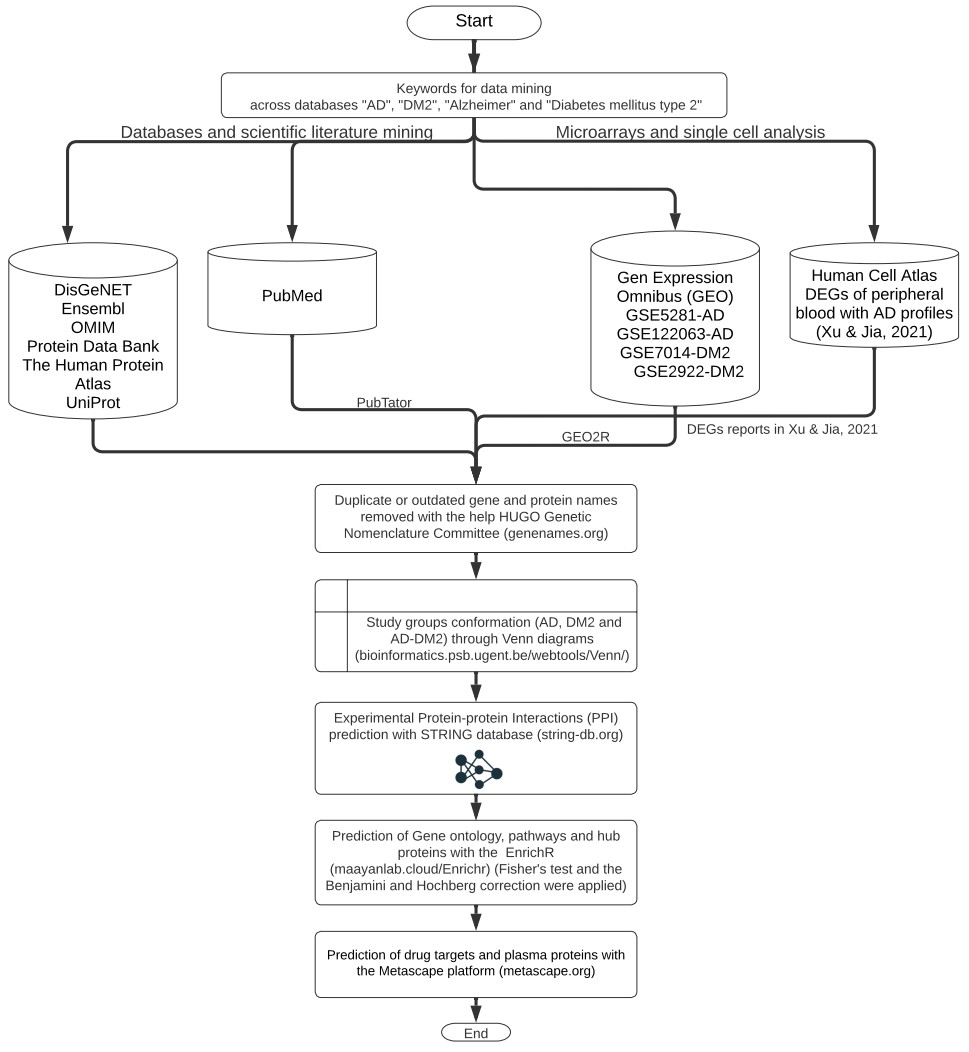

**Figure 1 The workflow of the methodology used.**

experimental evidence which include only biochemical or biophysical data produced from yeast two-hybrid experiments, and a minimum required interaction score >0.900, which represents the approximate probability that a link exits between two proteins in the same metabolic map in the KEEG database to reduce the rate of false positives (*Timalsina, Charles & Mondal, 2014*; *Szklarczyk et al., 2019*). Nodes that did not have a connection in the PPI were discarded.

## PPI Gene Ontology (GO) and pathway enrichment analysis

The edge lists of the PPI file were submitted to Gene Ontology Biological Process (GOBP) and KEGG 2021 pathway analysis through the EnrichR platform (https://maayanlab.cloud/Enrichr/) (*Chen et al., 2013*). These analyses were carried out for each of the disease study groups with an BHC adjusted value of $p < 0.05$ derived from a Fisher's

**Table 1** Total genes and proteins names mined by database for AD and DM2.

|  | DisGeNET | Ensembl | OMIN | Protein atlas | PDB | Uniprot | Gen expression omnibus | Cell atlas | PubMed |
|---|---|---|---|---|---|---|---|---|---|
| AD | 3342 | 37 | 185 | 107 | 86 | 85 | 221 | 436 | 77 |
| DM2 | 2727 | 77 | 177 | 20 | 204 | 237 | 26 |  |  |
| Total | 6069 | 114 | 362 | 127 | 290 | 322 | 247 | 436 | 77 |

exact test, and additionally to exclude the false positives associated with gene category enrichment analysis (*Fulcher, Arnatkeviciute & Fornito, 2021*), only the ten most significant GOBP and pathway terms were retained for analysis.

### PPI hub analysis

To determine the proteins of high biological value within the PPI networks groups (hub proteins), the hub option from the EnrichR platform (https://maayanlab.cloud/Enrichr/) was used. The Expression2Kinases program (*Clarke et al., 2018*) was used to identify regulatory proteins (mainly transcription factors (TFs) and kinases) involved in important signaling pathways that potentially regulate a PPI network based on the gene list submitted (*Chen et al., 2012*). Only proteins with adjusted $p < 0.05$ were considered significant, and the 10 most significant proteins for each study group (AD, DM2, and AD –DM2 association) were taken for analysis, as previously did for GOBP.

### Prediction of biomarkers and drug targets

To determine if the hub proteins could be possible biomarkers or drug targets, the protein names were submitted to the Metascape platform to match our data with the available options "plasma" (protein atlas) and "drug bank" (*Zhou et al., 2019*). The obtained data was plotted as a drug-target network with the Cytoscape platform v 3.9.0 (https://apps.cytoscape.org/).

## RESULTS

### A plenty of biological information shared between AD and DM2

To identify data shared by both AD and DM2, nine databases were mined (Table S1). After eliminating duplicate gene and protein name records in the databases (Table S2), most of the information available for AD and DM2 was from the DisGeNET ($n = 6,069$) and the information in the scientific literature (PubMed) was scarce ($n = 77$) (Table 1). Venn diagram analysis showed that data for AD was much more abundant than for DM2 (Fig. 2), but both diseases shared considerable biological information ($n = 1,551$). The shared data was considered as the AD-DM2 association group.

### Molecular complexity revealed by Protein-Protein Interaction (PPI) network analysis

The gene lists for the AD, DM2 and AD-DM2 groups were assigned to PPI in humans on the STRING platform (Table S3). Because only experimental information was considered, the number of interacting nodes in the predicted PPI networks was reduced but significant

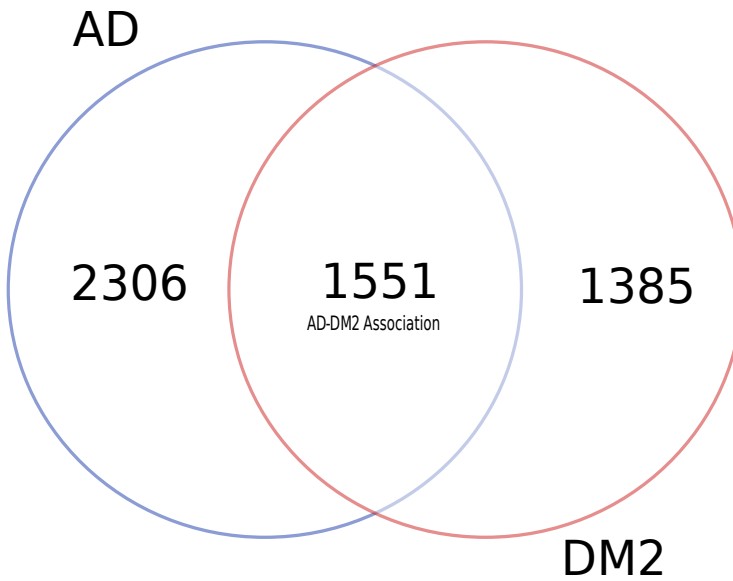

AD

2306    1551    1385
AD-DM2 Association

DM2

**Figure 2** **Distribution of unique and shared number of gene and protein information for AD and DM2.**

(Adj. p value < 0.05). The whole networks for each group of analysis showed similar interconnections (Figs. 3A–3C). The highest average connections were for the DM2 (average node degree = 2.97), followed by AD (average node degree = 2.35), and the AD-DM2 association (average node degree = 2.03).

## Shared biological process and pathways between AD and DM2 and specific to each of them

To understand the biological significance of the PPI data, analysis of GOBP and signaling pathways were performed for each group. The most significant (Adj. p value < 0.05) GOBP associated with AD was related to the "nuclear-transcribed mRNA catabolic process (GO:0000184)", followed by transcriptional and translation processes related to the neuron's cells. On the other hand, GOBP for DM2 were associated with "mitochondrial respiratory chain complex I assembly (GO:0032981)", followed by related bioenergetics terms. For DM2 the most significant pathway was "Thermogenesis". The GOBP analysis for the AD-DM2 association was related to the "cellular response to cytokine stimulus (GO:0071345)", followed by the "cytokine-mediated signaling pathway (GO:0019221)" that connects all group information (Fig. 4A). The signaling pathways for the AD-DM2 association were "pathways in cancer" "PI3K-Akt signaling pathway", and "Lipid and atherosclerosis" (Figs. 4B, Table S4). AD and DM2 shared pathways related to neurological diseases such as prion disease, Parkinson's, and amyotrophic lateral sclerosis.

## The main proteins (hubs) connecting AD and DM2

Hub analysis was performed to identify the main proteins that interact within the PPI networks. EnrichR showed SRC (a tyrosine-protein kinase) shared with the DM2 and

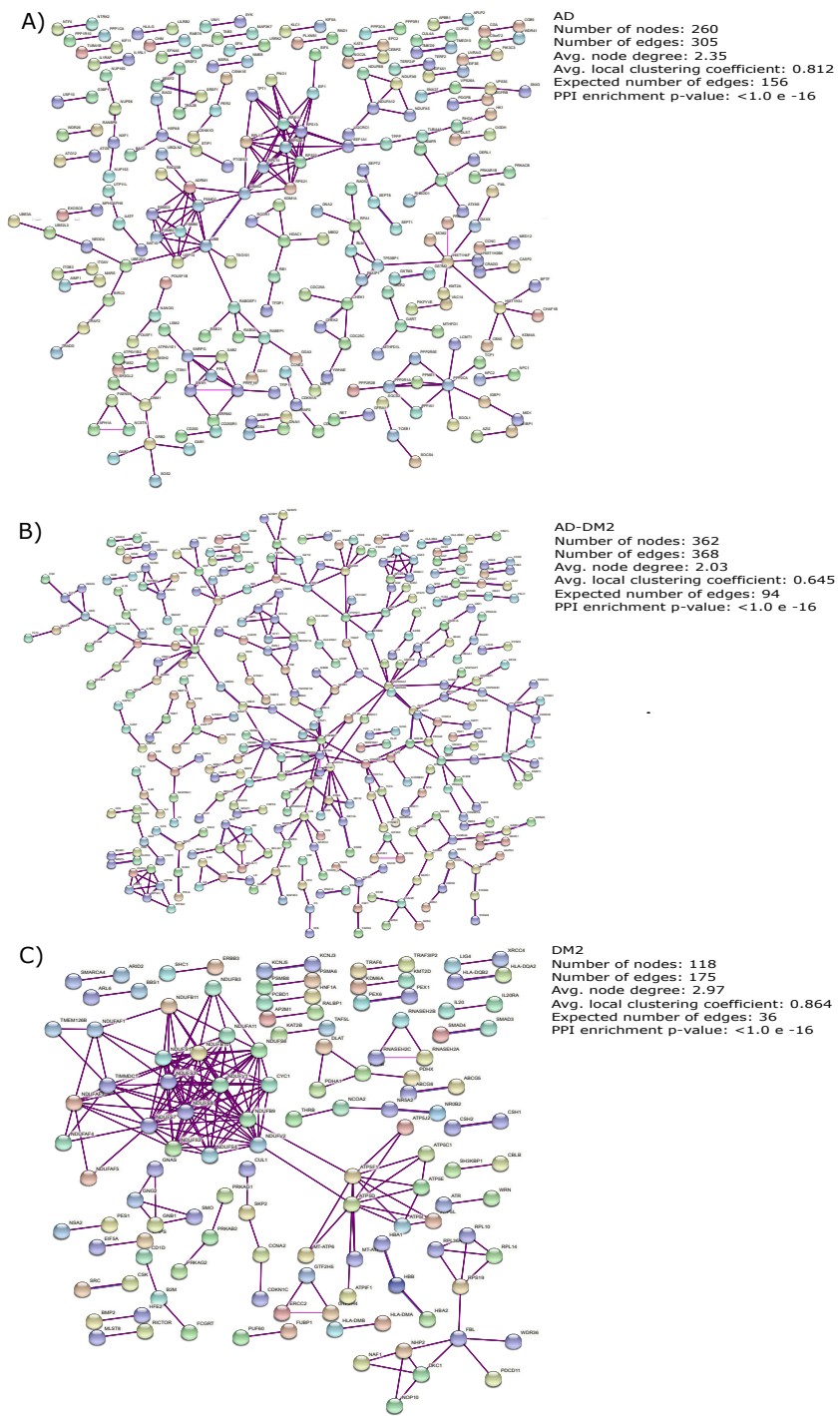

**Figure 3 PPI networks for the study groups.** The PPI images were produced on the string platform and the panels created on Inkscape (https://inkscape.org/).

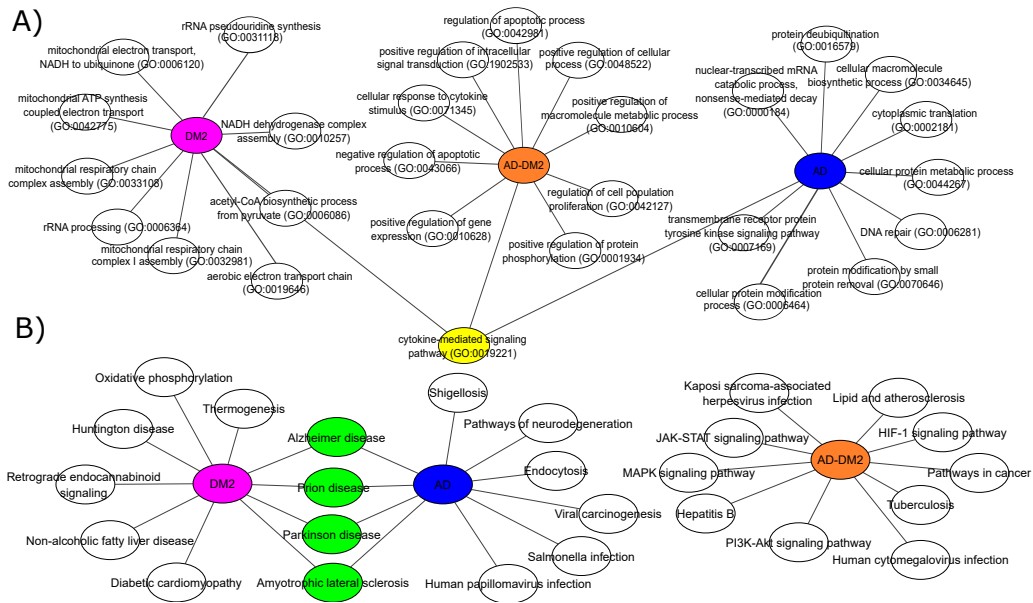

**Figure 4** **Biological processes (A) and signaling pathways (B) for the study groups AD (in blue), DM2 (in purple), and AD-DM2 (in orange).** In yellow and green are shown the shared ontologies and pathways respectively.

AD-DM2 data (Fig. 5, Table S5). Two proteins involved in the cell cycle proliferation (BRCA1) and a glucose transporter (SLC2A4) were shared between DM2 and AD. ESR1, a receptor of estrogens located at the center of the hub network, was shared with the three groups of study, underlining its potential importance.

## Potential biomarkers and drug targets for the AD-DM2 association

The Metascape platform showed that of the ten hub proteins identified in the association AD-DM2, seven (STAT3, EGFR, IRS1, MAPK1, SRC, HSP90AA1, PIK3R1) were matched as proteins present in plasma, except UBC, MAPK3, and ESR1. All of them could be inhibited by multiple drugs (Fig. 6). For example, EGFR is targeted by various monoclonal antibodies (mAbs). STAT3, SRC, and HSP90AA1 are targeted by several anticancer agents. HSP90AA1 is targeted also by the flavonoid Quercetin (DB04216), which also targets the ESR1 protein, which is not present in plasma but is shared by the study groups. This could be a novel drug target option to direct further treatments for AD-DM2 comorbidity. Those plasma-predicted proteins could also be good candidates to evaluate as biomarkers in the AD-DM2 association (Fig. 6, Table S6).

## DISCUSSION

AD and DM2 are complex diseases for which a link have been suggested (*Michailidis et al., 2022*). The large amount of data available for these diseases can be explored to identify novel patterns that could explain its pathogenic relation. In this work, through a data mining strategy of several databases, we found gene and protein information shared by
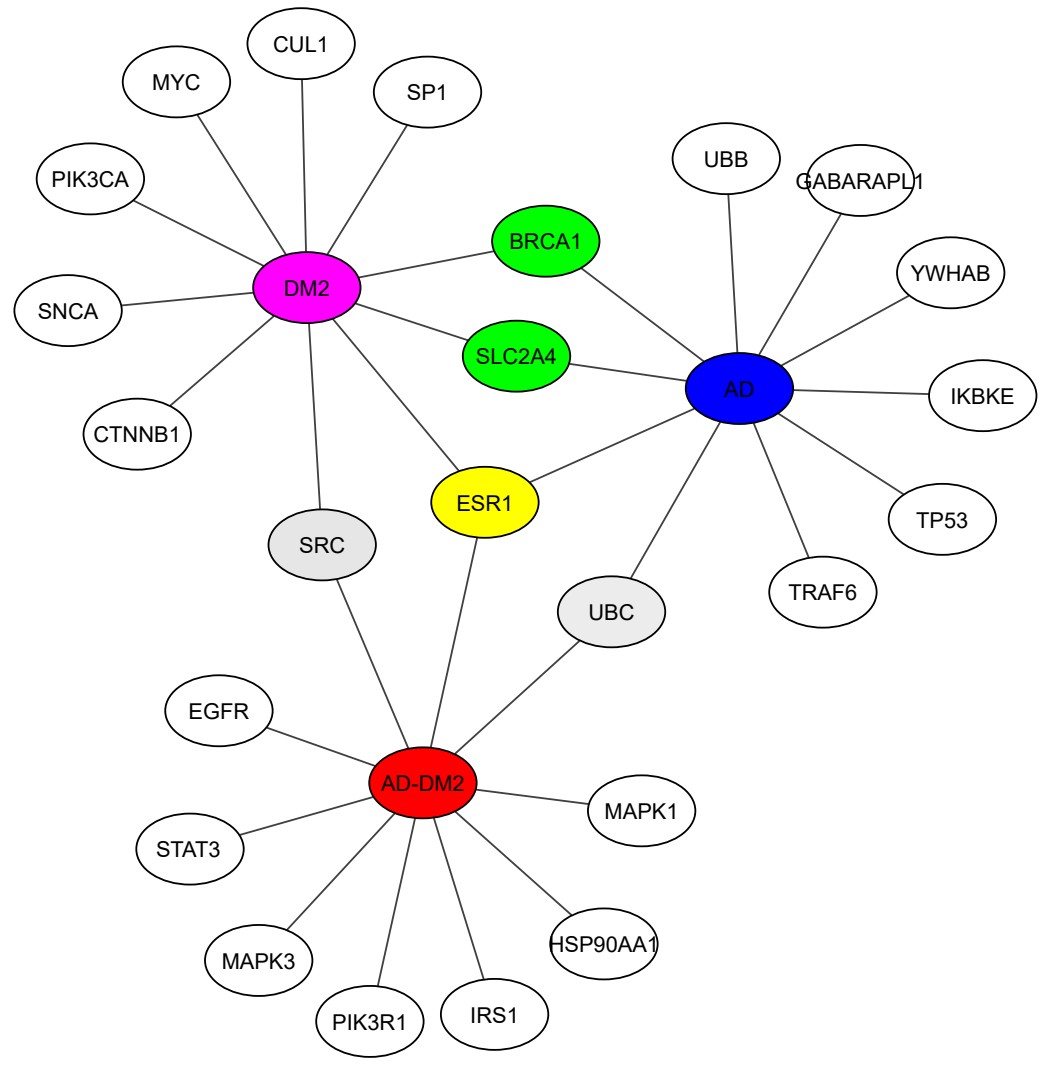

**Figure 5 Hub proteins identified by EnrichR.** In gray are shown the shared protein between AD-DM2 and DM2. In yellow is the protein shared by the three groups. In green the proteins shared by AD and DM2 The study groups are distinguished by specific colors AD (blue), DM2 (purple), and AD-DM2 (orange).

AD and DM2, demonstrating that there are potential molecular links for both diseases. A recent bioinformatic study based on gene expression data sets for AD and DM2 identified 241 deregulated in common for both diseases that could be implicated in the pathogenesis of its association (*Chung & Lee, 2021*). In contrast to the previous study, we focus our research on the integration of multiple sources of biological information for AD and DM2, which was then assigned to PPI experimental confirmed data on humans, in an effort to reduce the inclusion of false positives associated with this type of studies (*Mahdavi & Lin, 2007*). To better understand this information, we studied these diseases by considering three conditions: AD, DM2, and the AD-DM2 association. The networks were similarly interconnected in each study group, revealing their molecular complexity, especially for

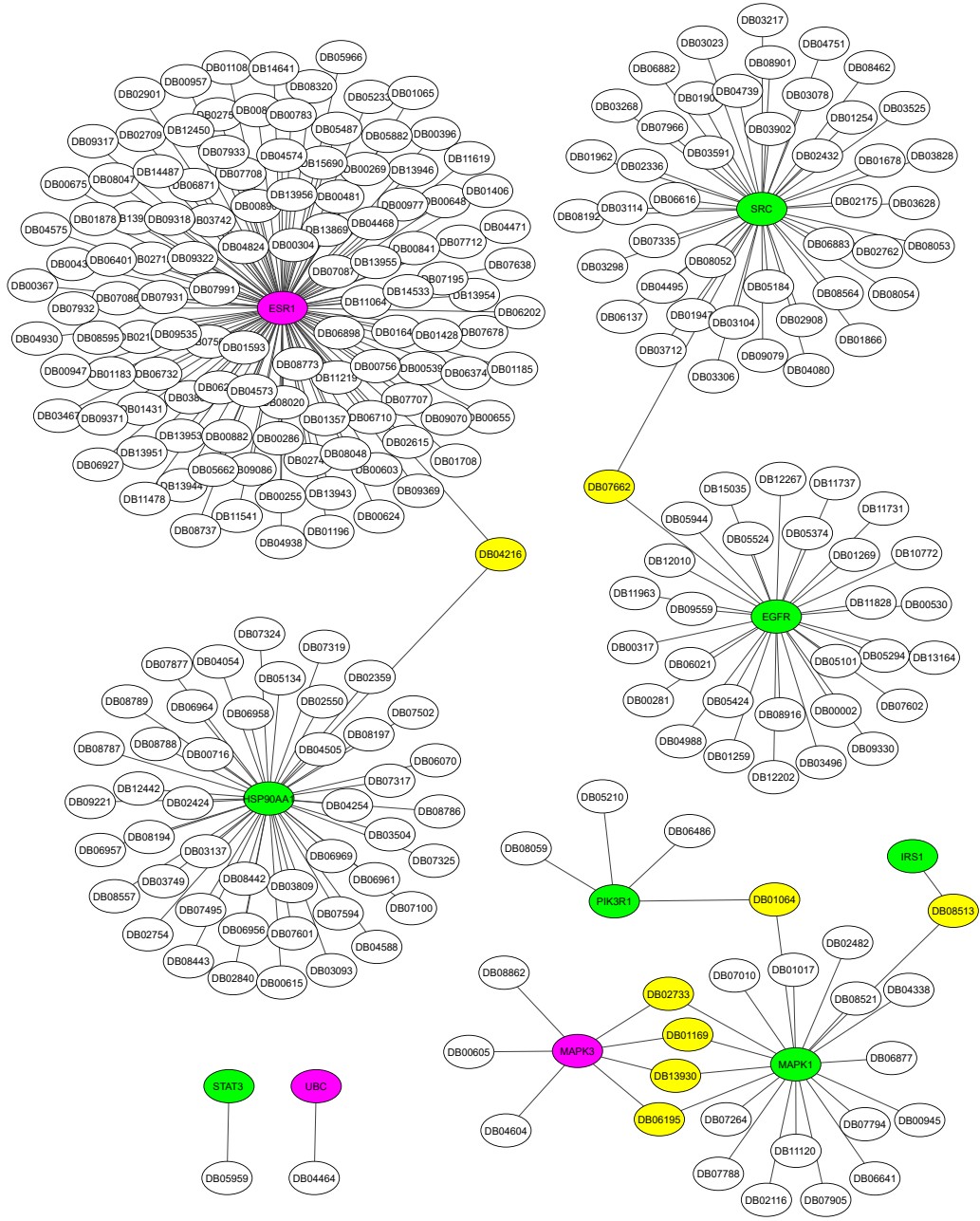

**Figure 6** **Drug-target network for hub proteins.** In green are shown the plasma hub proteins, in magenta the non-plasma proteins, and in yellow the shared drugs labeled with its drug bank ID. The whole data is in Table S6.

the DM2 (Fig. 3). This complexity could be explained by its specific GOBP and pathway deregulations.

Functional analyses based on GOBP for AD were found related to specific terms that affect the vital genetics processes of the cells. For example, the most significant were transcriptional and translation terms, which were also observed in gene expression
studies specific to AD (*Xue et al., 2020*), indicating that our method of multiple data integration is proper. In an epidemiological and pathological context, it is known that there are two main types of AD (*Tanzi, 2012*; *Reitz & Mayeux, 2014*). Early-onset Alzheimer's disease (EOAD) is the least common, with an estimated incidence of 10 cases per 100,000 people, affecting < 65 years of age, and associated with hereditary genetic factors (*Alzheimer's Association, 2016*; *Cacace, Sleegers & Van Broeckhoven, 2016*). As our analysis showed specific genetic terms, it suggests a logic connection with the genetic fact for the EOAD. On the other hand, late-onset Alzheimer's disease (LOAD) is the most common type in the world, appears at older ages (>65 years) and is not related to hereditary alterations (*Del Ser et al., 2001*). Meanwhile, the prevalence of DM at ages > 65 is higher (*Laiteerapong & Huang, 2018*), and in this age stage cognitive impairment is also known to start (*van den Berg et al., 2005*). In the same logic, GOBPs for DM2 were found to be restricted to the mitochondrial bioenergetics, which is known to be altered in this disease (*Pinti et al., 2019*). Interesting mitochondrial alterations in neurons are a consequence of DM2 (*Sato & Morishita, 2014*). Then again, the AD-DM2 association shares cytokine deregulation that connects both diseases (AD and DM2), suggesting that inflammation is the potential link. Indeed, evidence showed that in the pathophysiology of AD and DM2 the dysregulation of inflammation plays a key role (*de Nazareth, 2017*). Neuroinflammation produces accumulation of β-amyloids and consequently the release of cytokines by the activation of microglial cells (*Rosenberg, 2005*; *Michailidis et al., 2022*). Likewise, in DM2 are evidence of accumulation of amyloid beta protein (Aβ) and hyperphosphorylated tau protein in pancreatic tissue (*Miklossy et al., 2010*), a similar histopathological feature occurring in AD brains. Thus, if these two pathologies connect at some point, it could be due to that DM2 pathological processes lead in the long term to LOAD. Further longitudinal studies may clarify whether these molecular mechanisms are associated with the AD-DM2 link.

The hub proteins within the networks showed that some TFs, kinases, and ubiquitin proteins are shared between the study groups. Among the most relevant were UBC, SRC, ESR1, BRCA1, and SLC2A4. UBC, also known as ubiquitin C, has been associated with protein ubiquitination processes (*Zheng & Shabek, 2017*). This protein in patients with DM2 causes an accumulation of polyubiquitinated proteins in pancreatic tissue and leads to the apoptosis process, probably due to an increase in islet amyloid polypeptide (IAPP) oligomers (*Bishoyi et al., 2021*). In AD, there is also an accumulation of polyubiquitinated proteins similar to IAPP that leads to neuronal apoptosis and can potentially cross the blood–brain barrier, contributing to the development of AD (*Qosa et al., 2014*). SRC, a kinase, has been proposed as a therapeutic target in AD and DM2 (*Taniguchi et al., 2013*; *Beirute-Herrera et al., 2020*). ESR1, which is located in the center of the hub network map (Fig. 5), is an estrogenic receptor with risk polymorphisms for AD and DM (*Elcoroaristizabal Martín et al., 2011*; *Yang et al., 2018*). In women, the decrease in estrogen levels during menopause is associated with AD (*Mosconi et al., 2021*). This is also of particular importance because two-thirds of the cases of most prevalent LOAD are related (*Rahman et al., 2019b*). The tumor suppressor gene BRCA1 related with breast cancer in women is also involved in the pathogenesis of AD due to its accumulation in the

brain (*Nakamura et al., 2020*). Information on the role of BRCA1 in DM2 is scarce, but is associated with an increased risk of developing DM2 (*Bordeleau et al., 2011*). SLC2A4 or GluT4 participates in cognitive impairment, its production is reduced due to insulin resistance that also occurs in DM2 (*McNay & Pearson-Leary, 2020*). Although the literature information for these proteins is disease-specific, the connection found in our data analysis approach could be helpful to direct the further in-depth experimental studies to confirm or reject their role in the association AD and DM2.

Among the hub proteins related to the AD-DM2 association, some of those were targets of drugs already in use. The EGFR protein is a target of mAbs. This class of immunotherapy has been proposed for the treatment of AD; for example, mAbs against the A $\beta$-amyloid has been used (*van Dyck, 2018*). In DM1 and DM2, this type of therapy has also been devised; however, such therapy is ineffective (*Ke et al., 2021*; *Heymsfield et al., 2021*; *Shi et al., 2022*). For example, the mAbs ganterumab and solanezumab failed to slow cognitive decline in AD patients on phase II/III trials (*Salloway et al., 2021*). Moreover, the recent editorial notice of concern about the $\beta$-amyloid as the cause of Alzheimer's (*Lesné et al., 2006*; *Piller, 2022*) obligates the quest for new drug targets. Consequently, our finding that potential drugs and targets that could be used in the treatment of AD and DM2 is valuable. For example, the targets STAT3 and the hypoxia inducible factor-1 $\alpha$ (HIF-1) found related to the AD-DM2 association (Figs. 4B, 5) are inhibited by the ENMD-1198 (DB05959) (Fig. 6), a microtubule inhibitor (*Moser et al., 2008*). In the pathophysiology of AD, the microtubule-associated proteins (MAP/Tau) play an important role (*Dehmelt & Halpain, 2005*). In relation to DM2, microtubules regulate insulin delivery to the membrane for secretion, and its function is altered by abnormal glucose levels (*Trogden et al., 2019*). As microtubules participate in important physiologic aspects of AD and DM2, they are considered potential pharmacological targets (*Varidaki, Hong & Coffey, 2018*; *Ho et al., 2020*). Another interesting example is the flavonoid quercetin, a phytochemical found in diets of fruits and vegetables, that have shown neuroprotective effects against AD (*Khan et al., 2019*) as well as antidiabetic effects (*Eid & Haddad, 2017*). Therefore, since these identified drugs and targets are already being used, they could be used in drug repurposing efforts to guide the rational search for disease-modifying treatments for the AD-DM2 association. Additionally, the seven proteins could also be potential biomarkers, because they were predicted to be plasma proteins.

## CONCLUSIONS

In conclusion, we found considerable biological information that links AD and DM2. Prediction of PPI guided the inference of the potential dysregulated GOBP and pathways shared for both diseases or specific to each of them, highlighting the inflammatory deregulation for the AD-DM2 association. Analysis of hub proteins allows the identification of anticancer drugs and flavonoid nutraceuticals already in use, underlining potential drugs and targets for further drug repurposing efforts. In addition, those hub plasma-predicted proteins could be potential blood biomarkers that could lead to improved diagnostic strategies. Also, our data mining strategy to study the complex interactions underlying AD

and DM2 could be adapted to other diseases where an epidemiological or molecular link has been recognized.

### Funding
This research was funded by CONACyT, grant number 320004. The funders had no role in study design, data collection and analysis, decision to publish, or preparation of the manuscript.

### Grant Disclosures
The following grant information was disclosed by the authors:
CONACyT: 320004.

### Competing Interests
The authors declare there are no competing interests.

### Author Contributions
- Ricardo Castillo-Velázquez conceived and designed the experiments, performed the experiments, analyzed the data, prepared figures and/or tables, authored or reviewed drafts of the article, and approved the final draft.
- Flavio Martínez-Morales conceived and designed the experiments, analyzed the data, authored or reviewed drafts of the article, and approved the final draft.
- Julio E. Castañeda-Delgado conceived and designed the experiments, analyzed the data, authored or reviewed drafts of the article, and approved the final draft.
- Mariana H. García-Hernández analyzed the data, authored or reviewed drafts of the article, and approved the final draft.
- Verónica Herrera-Mayorga performed the experiments, analyzed the data, authored or reviewed drafts of the article, and approved the final draft.
- Francisco A. Paredes-Sánchez performed the experiments, analyzed the data, authored or reviewed drafts of the article, and approved the final draft.
- Gildardo Rivera analyzed the data, authored or reviewed drafts of the article, and approved the final draft.
- Bruno Rivas-Santiago analyzed the data, authored or reviewed drafts of the article, and approved the final draft.
- Edgar E. Lara-Ramírez conceived and designed the experiments, performed the experiments, analyzed the data, prepared figures and/or tables, authored or reviewed drafts of the article, and approved the final draft.

### Data Availability
The raw data are available as a Supplemental File.

## Supplemental Information

Supplemental information for this article can be found online at http://dx.doi.org/10.7717/peerj.14738#supplemental-information.

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
