# Peer review of "Bioinformatic prediction of the molecular links between Alzheimer’s disease and diabetes mellitus"

_PeerJ, doi:10.7717/peerj.14738_

## Round 0.1 · original submission · Major Revisions

Dear authors. Please note the indications of the reviewers. Regards.

Reviewer 1 ·

Basic reporting

1 The review of related work is not sufficiently thorough and not sufficiently specific. The authors should cite the latest references if appropriate.
2 Figure 5 needs to be improved.
3 L160-L161 seems not clear. The authors should rephrase it.

Experimental design

1 The authors should add more details in key parts to make the results replicable. For example, what are the gene selection criteria in each database? What is their selection cutoff in each dataset? What are the references they used in PubMed?
2 In fact, the interaction between gene and gene is more complex. I don’t think L104-L105 is a reasonable or robust way to identify key genes or AD-DM2 associations.
3 To my knowledge, there are various filtering criteria to determine PPI in published articles. Is there any reference to support the criteria the authors used (minimum required interaction score>0.9000)? The issues also apply to their PPI hub analysis.
4 The authors should provide information on statistical tests they used throughout the manuscript.

Validity of the findings

1 The authors should provide the gene lists they filtered in excel file format.
2 The authors should provide the complete lists for Table S4.
3 Most results are more like descriptive reports. The authors should improve them.
4 The Discussion section should be improved to better reflect the quality of the work.

Reviewer 2 ·

Basic reporting

no comment

Experimental design

no comment

Validity of the findings

no comment

Additional comments

This is an interesting study and the authors have mined nine databases to identify information that overlap AD and DM2. gene information was mapped to protein-protein interaction (PPI) networks based on experimental data.gene ontology biological process (GOBP), and pathway analysis accomplished.The authors found that the pathway information for DM2 and the association AD-DM2 were linked mainly to inflammatory alterations. 10 hub proteins were identified, and seven were predicted to be present in plasma.

It is a very interesting work. However, some major concerns should be addressed:

Abstract
- background, clarify the aim of this study.
- conclusion, Please focus the abstract on your study and your results.

Methods
- line 131, “Only proteins with p <0.05 were considered significant, taking only the 10 most significant proteins for each study group (AD, DM2, and AD DM2 association).” Why only top 10, please provide more literature support. And please demonstrate the number of all significant proteins if possible.

Result
- Line 158, “PPI manages important biological processes, including metabolism control (Rao et al., 2014).” That sentence should be in Methods.

Discussion
- “ESR1 is an estrogenic receptor with risk polymorphisms for AD and DM.” ESR1 is at the center of the hub, should be more thoroughly discussed.

Figure 2 & 3 please provide figures with better resolution.

Figure 5 is too complicated. Protein labels could have a smaller font size for better review

Reviewer 3 ·

Basic reporting

Hypothesis is clear but he overall methodology is not sufficiently provided
Figure are of low quality : I find it really difficult to draw any conclusions from the figures .
English need to be improved alot

Experimental design

Experimental section is not explained in detail needs to be improved

Validity of the findings

Since its an Insilco work , concluding as a major finding is not relevant or seems preliminary

---

## Round 0.2 · accepted · Accept

Congrats, after corrections, both reviewers recommend accepting your manuscript. Please, hold on to more instructions until the final publication of your manuscript.

Reviewer 1 ·

Basic reporting

My comments are addressed by the authors.

Experimental design

My comments are addressed by the authors.

Validity of the findings

My comments are addressed by the authors.

Additional comments

My comments are addressed by the authors, and the manuscript has been improved a lot.

Reviewer 2 ·

Basic reporting

no comment

Experimental design

no comment

Validity of the findings

no comment

Additional comments

Authors addressed comments well. I recommend this article be accepted.